# Dextro-Transposition of Great Arteries and Neurodevelopmental Outcomes: A Review of the Literature

**DOI:** 10.3390/children9040502

**Published:** 2022-04-02

**Authors:** Kalliopi Kordopati-Zilou, Theodoros Sergentanis, Panagiota Pervanidou, Danai Sofianou-Petraki, Konstantinos Panoulis, Nikolaos Vlahos, Makarios Eleftheriades

**Affiliations:** 1“Maternal-Fetal Medicine” MSc, 2nd Department of Obstetrics and Gynecology, Aretaieio Hospital, School of Medicine, National and Kapodistrian University of Athens, 11528 Athens, Greece; kpanoulis@med.uoa.gr (K.P.); nikosvlahos@med.uoa.gr (N.V.); melefth@med.uoa.gr (M.E.); 2Department of Hygiene, Epidemiology and Medical Statistics, School of Medicine, National and Kapodistrian University of Athens, 11528 Athens, Greece; tsergent@med.uoa.gr; 3Laboratory of Developmental Psychophysiology and Stress Research, Unit of Developmental and Behavioral Pediatrics, First Department of Pediatrics, “Aghia Sophia” Children’s Hospital, School of Medicine, National and Kapodistrian University of Athens, 11527 Athens, Greece; ppervanid@med.uoa.gr; 4First Department of Pediatrics, “Aghia Sophia” Children’s Hospital, School of Medicine, National and Kapodistrian University of Athens, 11527 Athens, Greece; danaisofianoupetraki@gmail.com

**Keywords:** dextro-transposition of great arteries, arterial switch operation, neurodevelopment, neurodevelopmental disorders

## Abstract

Background: Arterial switch operation (ASO) is the gold-standard surgical approach for dextro-transposition of the great arteries (D-TGA). It is performed during the neonatal period and has almost diminished the previously high mortality rate (from 90% if left untreated to <0.5%). Despite the impressively high survival rates, the surgical procedure itself—along with the chronic post-operative complications and the perinatal impaired cerebral oxygen delivery—introduces multiple and cumulative risk factors for neurodevelopmental impairment. Method: This study is a review of English articles, using PUBMED and applying the following search terms, “transposition of the great arteries”, “neurodevelopment”, “autism”, “cerebral palsy”, and “attention-deficit hyperactivity disorder”. Data were extracted by two authors. Results: Even though general IQ is mainly found within the normal range, D-TGA children and adolescents display reduced performance in the assignments of executive functions, fine motor functions, attention, working memory, visual–spatial skills, and higher-order language skills. Moreover, D-TGA survivors may eventually struggle with inferior academic achievements and psychiatric disorders such as depression, anxiety, and ADHD. Conclusions: The existing literature concerning the neurodevelopment of D-TGA patients suggests impairment occurring during their lifespan. These findings underline the importance of close developmental surveillance so that D-TGA patients can better reach their full potential.

## 1. Rationale

Congenital heart defects (CHD) are the most frequent congenital anomalies, affecting millions of newborns every year. The incidence of severe CHD that require intensive medical/surgical intervention is nearly 2.5–3/1.000 live births [1]. Among this group, dextro-transposition of the great arteries (D-TGA) accounts for approximately 3% [2]. The gold-standard surgical approach of D-TGA is the arterial switch operation (ASO), which has decreased the mortality rate from 90% for unoperated patients to <0.5% for patients who undergo corrective surgery [3]. However, considerable adverse consequences arise, encompassing repercussions outside the cardiovascular system. Critical CHDs, such as TGA, have been related to neurodevelopmental (ND) impairments in the affected children, who experience varying degrees of morbidities within most ND domains that can be established during childhood and which persist throughout adult life [4,5,6,7,8,9].

A great effort has been made to clarify the nature of ND impairments in CHD patients. Academic underachievement, non-typical or delayed development, and behavioral adversities are morbidities found in this population [8]. Moreover, it is suggested that CHD children are at greater risk of social-cognitive and social-communication deficits as well as autism spectrum disorder [10,11]. There is a variety of factors that accumulate from the prenatal period until adulthood that have been associated with adverse ND outcomes of CHD survivors, including TGA population. Firstly, intraoperative conditions that lead to cerebral perfusion abnormalities have been suggested as the primary mechanism of ND development [12,13,14]. Growing evidence supports that anatomical and functional ND impairments are also present before surgery [7,15,16,17,18,19]. Specifically, MRI findings have indicated mild ischemic lesions, in the form of white matter injury similar to periventricular leukomalacia, on CHD neonates pre-operatively [18,20,21]. These findings have led to the last hypothesis that abnormal brain development and adverse ND outcomes derive from events in utero. Even though CHD fetuses present “brain-sparing” autoregulation, they may have lower cerebroplacental ratio (CPR) and middle cerebral artery (MCA) pulsatility index (PI) [22,23,24,25], which means impaired oxygen distribution to the brain, and progressively smaller gestational age- and fetal-weight-adjusted total brain volume throughout the third trimester [26].

## 2. Objectives

CHD group is typically reviewed as an entity, and in most cases, results are presented as a whole for this heterogeneous group of cardiac defects. Therefore, limited data exist regarding the impact of each CHD on ND. This review aims to examine the association of TGA on adverse ND outcomes.

A growing number of studies attempt to assess the performance of TGA survivors in each ND domain. Different validated scales are used mainly according to the region that the study is performed and the ND domain assessed. Furthermore, each study focuses on a specific age population. The heterogenous information concerning ND development of TGA survivors is not accurately stated. Thus, this review was performed in order to gather and summarize all the existing data and clarify which ND domains are found impaired and the manifestations within different age groups. In order to achieve that, we addressed the following questions, according to PICO: (1) What are the TGA survivors’ neurodevelopmental impairments in different age groups? (2) Which validated neurodevelopmental (ND) scales are used? (3) Is there a discrepancy in the performance among TGA survivors and healthy peers? (4) Which particular domains are affected and what is the severity of the effect?

## 3. Materials and Methods

### 3.1. Eligibility Criteria

Population: D-TGA survivors of any age, with or without ventricular septal defect (VSD) or complex TGA, who were operated (ASO) in the neonatal period with deep hypothermia either with predominantly low-flow cardiopulmonary bypass (LFCPB) or with predominantly total circulatory arrest (TCA).

Intervention: Neurodevelopment of D-TGA survivors is evaluated by a variety of scales according to the neurodevelopmental domain assessed, age at time of the study, language, and region. Only studies using validated scales were enrolled in this review.

Outcomes: In order to be included, studies have to evaluate infants, children, or adolescents that had survived ASO, with a developmental scale appropriate for age, origin, and the developmental domains that needed to be assessed. These scales are interpreted by comparing the performance of patients to healthy peers. A study was excluded if it compared the scores of TGA patients to another group of CHD patients. Furthermore, trials that used other methods of ND evaluation, such as imaging tools or physical examination tests, were also excluded. We applied English as the language to include a study in the review.

### 3.2. Information Sources and Search Strategy

PUBMED was searched electronically on March 2021 utilizing a synthesis of relevant keywords and word variants for “transposition of the great arteries”, “neurodevelopment”, “autism”, “cerebral palsy”, and “attention deficit hyperactivity disorder”. Reference lists of associated reviews and articles were searched by hand for additional reports. 

### 3.3. Selection Process

Firstly, two authors independently searched the database and reviewed titles and abstracts. Subsequently, the two researchers screened full texts in order to define whether a study meets the inclusion criteria established by the PICO approach. In case of disagreement during the process, consensus was reached by discussion.

### 3.4. Data Collection Process

Two reviewers studied the papers independently and summarized the data on extraction forms designed a priori regarding study characteristics, design, and outcome. A consensus was reached among the reviewers by discussion whenever inconsistencies were raised.

### 3.5. Data Items

Any measure for each developmental domain (executive function, cognition, and adaptive function, speech-language and motor function, or neuropsychiatric domain) was eligible for inclusion. The scales used in each study should have evidence of validity. No restrictions were imposed according to the time of follow-up. In studies presenting outcomes for various CHDs, TGA measurements were isolated. In studies reporting multiple results, measurements were listed according to the ND domain assessed.

### 3.6. Effect Measures

Values were reported as mean SD for continuous variables. Data were reported as numbers and percentages of children affected for categorical variables and outcomes. 

## 4. Results

The literature search provided 99 relative citations. Of these citations, 47 were excluded due to irrelevant titles or abstracts. No duplicates were discovered. Full manuscripts of 52 studies were retrieved. In the review, 24 studies were included according to the eligibility criteria. Of the 28 excluded studies, some used imaging tools or examination tests to evaluate neurodevelopment of TGA patients and others correlated the results among different CHD groups and not the mean population. Unpublished relevant studies were not retrieved (Figure A1).

Developmental outcome data were presented in a table by age group according to Erikson’s Stages of Psychosocial Development [27]: infants (birth to ≤18 months), pre-school (19 months–<5 years), school age (5–11 years), adolescents (12–18 years), and young adults (19–40 years) (Table A1). 


**Infants (birth to ≤18 months)**


McGrath et al. was the first one to publish outcomes concerning neurodevelopment in infants with D-TGA that survived surgery [28]. The evaluated population belonged to the Boston Circulatory Arrest Study (BCAS). The aforementioned trial is a longitudinal prospective study that followed up 171 D-TGA patients who underwent ASO, from infancy to adolescence. Thus, it has offered the population base for many studies concerning D-TGA survivors. In the McGarth et al. study, 135 patients were estimated with Bayley Scale of Infant Development (BSID) and Fagan Test of Infant Intelligence at one year of age. Twelve (9%) children had Mental Developmental Index (MDI) scores ≤ 84, 28 (21%) children had Psychomotor Developmental Index (PDI) scores ≤ 84, and 23 (23%) children had novelty preference scores < 53% [28].

Park et al. followed up with 10 infants at 11–13 months of age using BSID II. Five infants of the studied population were diagnosed with mild and one with severe mental development delay. Furthermore, three infants were found with mild and one with severe psychomotor development delay [29]. The small sample size should be noted.

Andropoulos et al. included 20 patients that returned for 12 months of neurodevelopmental follow-up [30]. The results illustrated that cognitive, language, and motor score means were 0.3 SD, 0.67 SD, and 0.5 SD above reference population norms, respectively. Performance on adaptive-behavioral parameters of the Bayley Scales III, that were retrieved from the parental questionnaire, was 0.1–0.5 SD below norms for all scores, except from the practical one [30]. 

On the contrary, Lim et al. when evaluating 24 D-TGA survivors at the age of 18 months with BSID III, did not detect any different scores between the study group and mean population [31]. 

Early childhood and pre-school (19 months–<5 years)

From the studies selected for this review, the oldest entry belongs to Mendoza et al. [32]. In this study, 24 patients at the age of 1 to 5 years old were assessed with the Bayley Scales of Infant Development, for children up to 2 years, the Stanford-Binet Intelligence Scale and Denver Developmental Screening Test, for patients older than 2 years. The neurodevelopmental follow-up exhibited normal scores (≥84 or <1 SD below the mean) in 18 (75%) of the patients. Only three patients were estimated as dubious, with scores ranging from 68 to 83 (between 1 and 2 SDs below the mean). Additionally, three had scores below 68 (>2 SDs below the mean) and were considered to have abnormal neurodevelopment.

Similarly, all included studies concerning this age group [33,34,35,36], except Bellinger et al. [37], demonstrated that the mean IQ of D-TGA survivors was within the normal range. As far as visual–motor integration is concerned, Bellinger et al. and Ellerbeck et al., contrary to Hövels-Gürich et al. and Neufeld et al., suggested that it differed significantly between the patients and the mean population of the same age [33,34,36,37]. In the Bellinger et al. study, expressive language, motor function, and oromotor control were significantly affected [37]. Contrariwise, three other studies in this age group indicated that complete developmental scores did not vary from those of healthy peers [33,35,36]. 

School age (5–11 years)

The studies reviewed for this specific age group concluded in the same outcome. General IQ of school-age children with TGA is not found decreased compared to the mean population of the same age. Hövels-Gürich et al. presented midterm results of cognitive and motor development in children at a mean age of 5.4 years and 10.5 years after neonatal ASO performed between 1986 and 1992 with combined DHCA and low-flow CP, respectively [38,39]. They indicated reductions in vocabulary, fine and gross motor functions, and acquired abilities. 

In the same way, Calderon et al. conducted two different studies with children of school age in order to assess executive functions (cognitive and response inhibition, verbal and spatial working memory, and planning) [40,41]. It became evident that D-TGA survivors carried significant impairments in inhibition and cognitive flexibility despite normal working memory.

Karl et al. identified an increased risk of parent-reported psychosocial maladjustment in children with D-TGA, as well as teacher-reported various speech and expressive language problems and minor behavioral problems [42]. 

Bellinger et al., at eight years assessment of the BCAS, identified that mean scores of most ND scales were within the normal range [43]. However, neurodevelopmental status was found impaired in many aspects, including academic achievement, working memory, hypothesis-generating, sustained attention, fine motor function, visual–spatial skills, testing, and higher-order language skills.

Adolescents (12–18 years)

Eight of the 24 studies included in this review focused on the neurodevelopmental outcomes during adolescence [44,45,46,47,48,49,50,51]. The main domains of interest appeared to be psychosocial functioning and cognition of D-TGA survivors. Specifically, three of the studies illustrated that these adolescents are at greater risk of developing symptoms within the clinical range of attention-deficit/hyperactivity disorder (ADHD) during their lifespan, although the difference to the mean population was not always statistically significant [45,50,51]. In addition, inconsistent results were reported when using parent- and self-reports tools for psychiatric symptoms’ assessment [45,46]. Demaso et al. revealed that depressive, anxiety, and post-traumatic stress symptoms were more frequent within the D-TGA group, whereas Heinrichs et al. suggested significantly reduced psychosocial distress level, as more than two-thirds of the patients estimated their condition within the normal range. 

Regarding cognition, deficits in overall memory abilities were illustrated [44,47,49]. General intelligence of TGA adolescents was not found impaired compared to population norms [46,49]. Regarding academic attainments, Muñoz-López et al., contrary to Bellinger et al., reported that they were within the normal range [44,49]. Focusing on executive function, greater odds of impairment were outlined in letter fluency [48,49]. 

Young adults (19–40 years)

While searching for studies dealing with neurodevelopment in young adulthood, only one emerged [52]. Kasmi et al. explored neurocognitive and psychiatric outcomes in D-TGA patients corrected by ASO, using a variety of tests for both domains. Adults with D-TGA displayed significantly poorer performances in tasks assessing memory, visual–spatial skills, executive functions, and attention (all *p* < 0.05). Furthermore, patients presented increased prevalence of depression (*p* = 0.008) and anxiety disorders (*p* = 0.025) throughout their lifetime.

## 5. Discussion

The findings of this review regarding the first age group suggest that D-TGA infants who survived ASO do not consistently present neurodevelopmental impairment. We have demonstrated that most studies regarding this age group reveal both Mental Developmental Index (MDI) and Psychomotor Developmental Index (PDI) to fall within the normal range [28,30,31]. Only one of the four studies identified lower MDI and PDI scores, but no statistical significance was mentioned [29]. The small size of the cohort in most cases should be noted since it confines the strength of the results [29,30,31]. Homogeneity in the assessment scale was observed only in this particular age group. All studies evaluated patients with the Bayley Scale of Infant Development. 

The authors attempted to associate specific parameters to lower ND scores. Andropoulos et al. suggested that risk factors include preoperative MRI brain injury, preoperative and intraoperative cerebral oxygen saturation, total bypass time, and total midazolam dose [30]. Park et al. exploited proton magnetic resonance spectroscopy to measure the cerebral metabolism of TGA infants and attempted to correlate the abnormal findings to ND outcomes [29]. Lim et al. highlighted the negative impact of surgery beyond two weeks of age on brain growth and language development in infants with TGA [31]. Lastly, McGrath et al. tried to evaluate whether test scores of CHD patients in infancy are predictive of ND status at school age. They suggested that the association between test scores at one and eight years is modest and many children who are at risk of poor late outcomes will not be identified based on 1-year test scores [28]. 

Following the next age group, general IQ scores of pre-school D-TGA survivors were also mainly found within the normal limits. In this age group, we came across a great paradox. The study of Bellinger et al., which included the 158 children of the Boston Circulatory Arrest Trial, concluded significant impairments in the neurodevelopment of D-TGA survivors [37]. At the same time, most of the remaining studies did not meet statistical significance for the same neurodevelopmental domains [33,34,35]. The consensus was only met between Ellerbeck et al. and Bellinger et al. concerning language and visual–motor integration [34,37]. Neufeld et al. showed that the rate of autism in the group of pre-school children was unexpectedly high, roughly 10 times higher than current prevalence estimates of 6 per 1000 [36]. 

As far as risk factors are taken into account for the specific age group, familial predisposition should be taken into consideration [34,36]. Patients with low neurodevelopmental scores were prone to abnormalities on CT scans of the brain, head circumference less than the fifth percentile, and emergency switch operation or emergency balloon atrial septostomy [32]. Neonatal ASO with combined circulatory arrest and LFCPB was associated with neurological morbidities but not with decreased general IQ. Other potential modifiable variables constitute high plasma lactate levels preoperatively [36]. 

When estimating the results of school-aged children, someone expects to find consistency with the results of pre-school children, as many studies were conducted longitudinal in the same sample. Specifically, Hövels-Gürich, Bellinger, and Calderon examined the same group of children throughout their lifespan [38,39,40,41,43]. Bellinger’s results did not meet an agreement as he issued impaired general intelligence at four years of age and normal average IQ in the same children at eight years [35,40]. Calderon et al., focusing on executive function, displayed adverse scores in cognitive inhibition and cognitive flexibility, that persisted from ages 5 to 7, as well as frustration planning a strategy to achieve a goal, i.e., anticipating the correct number of actions to reproduce a visual model [40,41]. Executive function impairments seemed to have an early onset during the pre-school years. A significant proportion also struggled to determine the emotional and cognitive states of others (Theory of Mind deficits). Hövels-Gürich et al. demonstrated accordance among their results at 5.4 and 10.5 years [38,39]. Both studies concluded with findings of poorer motor functions, language skills, and acquired abilities within D-TGA patients. The rate of adverse outcomes at school age (55%) was twice the rate at age 5 (26%), a fact that supports the presumption that these children may straggle more as they grow older in particular abilities.

To our knowledge, adolescents with D-TGA constitute the most evaluated age group since we retrieved eight manuscripts concerning neurodevelopmental impairments that emerge during adolescence. Generally, lower than anticipated scores were found in academic achievements, visuo-spatial skills, memory, psychosocial, and executive functions. Two studies also identified a higher rate of internalizing (i.e., anxiety, somatic complaints, depressive symptoms) and externalizing problems on both parent and self-report measures [44,45]. 

Psychosocial and ADHD sequelae are a matter of great concern in CHD population in general. Even though no statistical significance was consistently met, it appears that TGA survivors should be closely monitored as they found themselves at risk of these conditions. Taking into consideration the BCAS, 16-year-olds with TGA were more prone than the control group (35% versus 20%) to a lifetime psychiatric diagnosis when assessed with the Schedule for Affective Disorders and Schizophrenia for School-aged Children (K-SADS) [44]. As for ADHD, it was suggested that adverse psychosocial status in adolescence was highly predicted by attention deficits at age 8 years [50]. Therefore, specially designed studies that focus on the treatment of childhood ADHD are needed in an effort to promote adolescents’ well-being. Demaso et al. proposed primary care clinicians and cardiologists consider psychotropic medication as a treatment option, given the high rates correlated to D-TGA, the efficacy of stimulants, and the reported tolerability of these medications in children with CHD [45]. 

Efforts to identify prognostic factors of ND impairments have also been identified in these age group studies. Specifically, while extracting the results from Muñoz-López et al., we came across the attempt to correlate the degree of memory impairment to the degree of hippocampal atrophy [49]. Although cyanosis occurred in all of the children, only a small proportion had detectable hippocampal pathology. No other structure can fully compensate for hippocampal injury, even as early as neonatal life, and its unique role in serving episodic long-term memory is to be investigated. Moreover, according to Heinrichs et al. 32% of the D-TGA presented with moderate or severe structural brain abnormalities in magnetic resonance imaging [46]. Periventricular leukomalacia was demonstrated in >50% and the severity of the impairment was linked to the grade of neurologic morbidity. The last one was significantly correlated with decreased intelligence, orthography, and analytical thinking. 

The majority of the studies established explicit neuropsychological weaknesses in children with TGA and recommended the utilization of tailored programs of interventions. Academic problems, along with long-term neuropsychological sequelae, might be mitigated over a D-TGA patient’s lifetime by the incorporation of these programs. Continuous surveillance of D-TGA survivors, as well as further research on neurobiological and psychological mechanisms of impairment, are needed to help fill gaps in our knowledge.

Our study should be perceived along with its limitations. Most of them are induced by a possible selection bias. The studies included in this review present heterogeneity among cohort characteristics, assessment tools, age of TGA patients, and rate of follow-up. Furthermore, a discrepancy is found among the types of studies themselves, including cohort, case-control, and observational. We should underline the restricted strength of the observational studies. Moreover, many studies’ outdated operative/perioperative strategies constitute a confounding factor so that current cohorts of TGA survivors could benefit from advanced surgical and medical intervention.

## 6. Conclusions

The existing literature suggests neurodevelopmental impairment of D-TGA patients occurring during their lifespan. New advanced technologies may eliminate operative risks and achieve a better understanding of the underlying mechanisms. These data emphasize the importance of close developmental surveillance starting early in life in order to best help D-TGA patients to reach their full potential.

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
