# Peer review of "Dextro-Transposition of Great Arteries and Neurodevelopmental Outcomes: A Review of the Literature"

_children, 2022, doi:10.3390/children9040502_

Round 1

Reviewer 1 Report

Thank you for the opportunity to review “Dextro-Transposition of Great Arteries and Neurodevelopmental Outcomes: a Systematic Review”. 

The paper requires significantly improved setting out of the theoretical background, methods, and authors' material consolidation. 

Here are my specific comments as we run through the manuscript. 

TITLE

Please verify the adoption of the sentence “systematic review”

INTRODUCTION 

  1. The introduction lacks the theoretical background to understand the relevance of the four research questions. The authors cited only three references (Donofrio et al., 2011, Limperopoulos et al., 2010, Lynch et al., 2014) to justify the literature debate and uncertainty about this issue. In other words, I suggest clarifying the theoretical and practical problems that led the authors to approach the literature review. In my opinion, the introduction did not introduce the reader to the four research questions. 
  2. Please verify line 61: (Donofrio et al., 2011); (Limperopoulos et al., 2010); (Lynch et al., 2014).

METHODS 

  1. The author claimed a systematic review, but the criteria for systematic review are not satisfied. Please verify recommendations for systematic review in Page, M. J., McKenzie, J. E., Bossuyt, P. M., Boutron, I., Hoffmann, T. C., Mulrow, C. D., ... & Moher, D. (2021). The PRISMA 2020 statement: an updated guideline for reporting systematic reviews. BMJ, 372.
  2. The search strategy is not well introduced—for example, the reason for introducing the term “autism”. Even if literature reported the association between autism and CHD, they did not mention this association in the introduction, and the word appears new for the reader. 
  3. It is unclear if the research question has been framed into PICO (population, intervention, comparison, outcome) format. PICO format is introduced in the study selection procedure “Studies included in this systematic review were selected according to the following 83 criteria: population, outcome, and study design.”
  4. The author wrote “Selected patients did not have a history of severe prematurity or low birth weight, an associated extracardiac anomaly, previous cardiac surgery, and associated cardiovascular anomalies requiring aortic arch reconstruction or additional open surgical procedures”. It is not clear if the author are exclusion criteria. 
  5. I suggest rephrasing the sentence: Neurodevelopment of the population mentioned above was evaluated by a great variety of validated scales, according to the study region, the age of TGA survivors at the time of the study and the neurodevelopmental domain assessed. All different scales were enrolled in this review.”. Indeed, the sentence appeared to be a result rather than part of the method. 

I suggest re-organizing sentences in the different sections of the method. For example, the first sentence of the section “data extraction” is : “All records were reviewed independently by two authors. Initially, they searched the 100 database and retrieved full texts of the studies that met the inclusion criteria established according to the PICOS approach. Note that the section should explain the extraction data process and not selection studies. Also, the PICO approach mentioned is it not previously described. 

Please note criteria for systematic review in http://www.prisma-statement.org

RESULTS

FIGURE A1 is incomplete and it did not address the criteria for systematic review. Please see http://www.prisma-statement.org

TABLE 1: I suggest rephrasing the title of the table. Please verify if the sentence “The Boston Circulatory Arrest Study” is correct for population’s characteristics. 

It is unclear the meaning of presenting five tables rather than one table also organized for the age of participants. In other words, I suggest summarized data into one table presenting data also by age 

Please note line 177 “Calderon et al.” Please review the entire manuscript.

Some sentences are hard to read. For example, line 170 “The studies reviewed for this specific age group all concluded in the same outcome. General IQ of school-age children who suffered from D-TGA and were operated in the first year of life is not found decreased compared to the mean population of the same age. Please consider splitting it into two sentences. 

Please note references format at line 204. Regarding cognition, Bellinger et al., 2011; Cassidy et al., 2017; Muñoz-López et al., 204 2017, illustrated deficits in overall memory abilities. Heinrichs et al., 2014, alongside Muñoz-López et al., 2017, revealed that adolescents' general intelligence was not reduced compared to population norms. Muñoz-López et al., 2017, reported that academic attainments were at the same level as mean adolescents, in contrast to Bellinger et al. Focusing on executive function, Cassidy et al., 2015; Muñoz-López et al., 2017, outline greater odds of impairment in letter fluency.

DISCUSSION

At the beginning of the section the authors wrote “The findings of this systematic review suggest that D- TGA infants who survived ASO do not consistently present neurodevelopmental impairments. Additionally, all studies regarding this age group reveal that both Mental Developmental Index (MDI) and Psychomotor Developmental Index (PDI) fall within the normal limits.” 

Authors should be more hedging since they did not rely on a strong work methodology. The review is not performed according to appropriate criteria (or not clearly described and presented), and several limitations affect their research. This is not only a theoretical issue. Indeed, scientific reports should inform policy makers and health action and uncertainty should be clarified to avoid malpractice in the health system.

The discussion should be improved since it appears fragmented. Several sentences are not clear and should be rephrased or clarified. Here only two examples:

  • Another interesting outcome that derives from Neufeld et al., 2008, study is the unexpectedly high rate of autism in the sample of children undergoing the ASO, roughly ten times higher than current prevalence estimates of 6 per 1000. As far as risk factors are concerned for the specific age group, the familial tendency should be taken into consideration (Ellerbeck et al., 1998; Neufeld et al., 2008)Κάντε _κλικ _ή _πατήστε _εδÏŽ _για _να _εισαγάγετε_κείμενο._.
  • While extracting the results from Muñoz-López et al., 2017, we came across the attempt to correlate the degree of memory impairment to the degree of hippocampal atrophy. Although cyanosis occurred in all of the children, only a subgroup had detectable hippocampal pathology. Even as early as neonatal life, no other structure can fully compensate for hippocampal injury, and its unique role in serving episodic long-term memory is to be detected.

Reviewer 2 Report

The review article is written effectively. The authors have summarized the articles in table format. 

Reviewer 3 Report

Valuable review article

Round 2

Reviewer 1 Report

Dear Editor,
Thank you for the opportunity to review further the paper entitled "Dextro-Transposition of Great Arteries and Neurodevelopmental Outcomes: a Systematic Review". 
I really appreciate the efforts of the authors to improve the manuscript. However, based on several methodological concerns, I advise against publishing the manuscript as a systematic review.

Here are some specific comments. 

  • The authors added the sentence (line 97): "This study was performed according to a protocol designed a priori and registered to the International Prospective Register of Systematic Reviews (PROSPERO)." 

The authors did not report the ID identification. However, the protocol is available at the following link: https://www.crd.york.ac.uk/prospero/display_record.php?RecordID=306165
It should be noted that the authors submitted the PROSPERO protocol the 23 January 2022. The submitted protocol declared that data extraction, risk of bias assessment, and data analysis were not started. Am I right? 

  • The authors wrote (line 109) "Outcomes: In order to be included, studies have to evaluate infants, children, or adolescents that had survived ASO, with a neurodevelopmental scale appropriate for that fitted their age, origin, and the developmental ND domains needed to be assessed. These ND scales are interpreted by comparing the performance of patients to healthy peers or the mean population. A study was excluded if it compared the scores of TGA patients to another group of CHD patients. Furthermore, trials that used other methods of ND evaluation, such as imaging tools or physical examination tests, were also excluded. We applied English as the language to include a study in the review.

    The authors should carefully review this section since, in my opinion, they mixed information about outcomes, exclusion criteria, etc etc. 

  • The authors wrote (line 59) "Moreover, it is suggested that CHD children are at greater risk of social-cognitive and social-communication deficits as well as autism spectrum disorder (10,11)". 

    I appreciated the author's effort to introduce the term "autism" in the search strategy. In my opinion, the authors should clarify the rationale to include "ADHD" and exclude "intellectual disabilities" (according to DSM5, neurodevelopmental disorders include various diagnoses). 

    It should be noted that inappropriate or inadequate search strategies may fail to identify records that are included in bibliographic databases, please note The Methodological Expectations of Cochrane Intervention Reviews (MECIR) https://community.cochrane.org/sites/default/files/uploads/MECIR-February-2021.pdf 

  • The authors did not assess the risk of bias for included study. It is a critical point since the assessment of the risk of bias support the extent to which its findings can be believed. Please note that in PROSPERO the protocol declared "The quality of each study is evaluated according to the study design". Am I right?

  • The authors wrote, "PUBMED was searched electronically on March 2021". 

    In my opinion, this is also a critical point of the review because the authors approached only the PUBMED database. 

  • In the discussion (line 277), the authors wrote "ND scores". 
    Please clarify the meaning of this term 

  • Please review manuscripts for various language included. For examples line 349: Κάντε _κλικ _ή _πατήστε _εδÏŽ _για _να _εισαγάγετε _κείμενο;  line 356: Κάντε _κλικ _ή _πατήστε _εδÏŽ _για _να _εισαγάγετε _κείμενο._.

  • The authors wrote (line 336) "Psychosocial and ADHD sequelae are a matter of great concern in CHD population in general. Even though no statistical significance was consistently met, it appears that TGA survivors should be closely monitored as they found themselves at risk of these conditions morbidities.". 

    In my opinion, the authors should clarify the neurodevelopmental disorder intended with the term "psychosocial sequelae".

  • I appreciated the author's efforts to improve the table. However, in my opinion, it is still hard to read. 
